Visual codon: a user-friendly Python program for viewing and optimizing gene GC content

Lin Shiming 1
Xu Fei 2
Huang Bifang 3
Zhao Li-li 4
Pan Danni 2
http://orcid.org/0000-0001-7181-4261 Lin Shiqiang 3 linshiqiang@fafu.edu.cn
1 School of Computing and Information Science, Fuzhou Institute of Technology , Fuzhou, Fujian , China
2 College of Agronomy, Fujian Agriculture and Forestry University , Fuzhou, Fujian , China
3 Life Science College, Fujian Agriculture and Forestry University , Fuzhou, Fujian , China
4 National Key Laboratory of Intelligent Tracking and Forecasting for Infectious Diseases, National Institute for Communicable Disease Control and Prevention, Chinese Center for Disease Control and Prevention , Beijing , China
Brogna Saverio
Electronic publication date: 2024 Dec 20
Publication date: 2024
Volume: 12
Electronic Location ID: e18755
Received 2024 Oct 4; Accepted 2024 Dec 3
Copyright: © 2024 Lin et al.
Copyright year: 2024
Copyright holder: Lin et al.
License: This is an open access article distributed under the terms of the Creative Commons Attribution License, which permits unrestricted use, distribution, reproduction and adaptation in any medium and for any purpose provided that it is properly attributed. For attribution, the original author(s), title, publication source (PeerJ) and either DOI or URL of the article must be cited.
License URL: https://creativecommons.org/licenses/by/4.0/

Keywords: Codon optimization, GC content, Python, Subcloning, Gene synthesis

Funding: The authors received no funding for this work.

==============================
Due to the codon bias of different species, codon optimization is usually carried out in the process of heterologous protein expression. At present, there are a variety of codon optimization tools. However, the optimized sequences may still have high or low points of local guanine and cytosine (GC) content, which is not conducive to the primer design of gene subcloning, and also makes it difficult to perform the experiment of synthesizing the whole gene with DNA fragments by polymerase chain reaction (PCR) reaction. In this study, we present a stand-alone software written in Python, with which users can manually check and adjust the GC content of sequence-optimized genes. The software takes the codon frequency of Escherichia coli as default and can work with other species as well. It provides a Graphical User Interface (GUI) interface, which allows users to change codons and intuitively see the effect of codon changes on local GC content. Our program brings convenience for the optimization of gene GC content and the subsequent gene cloning experiments.

Introduction

In the process of protein synthesis, codons play an important role in translating genetic information in mRNA into protein. Different species may prefer to use different codons for the same amino acid. Although the natural causes of codon bias are not known, the impact of this phenomenon on protein expression efficiency is significant (Arella, Dilucca & Giansanti, 2021; Iriarte, Lamolle & Musto, 2021; Parvathy, Udayasuriyan & Bhadana, 2022). Practically, for the optimal expression of the recombinant protein, it is often necessary to optimize the gene sequence according to the codon preference of the expression host. In addition, codon optimization has other applications, such as improving polymerase chain reaction (PCR) amplification and DNA cloning efficiency by optimizing guanine and cytosine (GC) content and eliminating repetitive regions (Chilamkurthy et al., 2022; Li, Jiang & Lu, 2018). For codon optimization, commercial companies such as Thermo (GeneOptimizer), Integrated DNA Technologies (IDT) (Codon Optimization Tool), and Genscript (GenSmart), have developed their codon optimization systems (https://github.com/shiqiang-lin/visual_codon/blob/main/Supplementary%20Material%20S1.txt). Moreover, a series of codon optimization tools, such as ATGme (Daniel et al., 2015), Codon optimizer (Fuglsang, 2003), CodonWizard (Rehbein et al., 2019), Codon Optimization Strategy with Multiple Objectives (COSMO) (Taneda & Asai, 2020), OPTIMIZER (Puigbo et al., 2007), DNA Chisel (Zulkower & Rosser, 2020), GeneOptimizer (Raab et al., 2010), BaseBuddy (Schmidt et al., 2023), and Improving Codon Optimization with RNNs (ICOR) (Jain et al., 2023), have been developed to support the heterologous expression of proteins. Some programs such as DNA Chisel and BaseBuddy, let the user set bounds on local GC content (over a given window size) and optimize the sequence according to these constraints. However, the sequences optimized with some commercial tools such as IDT (Codon Optimization Tool) and Genscript (GenSmart), may still have some regions where the GC contents are too high or too low. These commercial software solutions do not provide a graphical view of the local GC content of the gene sequence after optimization. In this case, it is often necessary but inconvenient for the users to manually change some codons on the basis of the optimized sequence in order to proceed with the experimental design, such as the synthesis of the entire gene with DNA fragments in a PCR reaction (Hu et al., 2022; Li, Liang & Qi, 2004; Zhao et al., 2022).

PCR is an important molecular biology technique used to amplify a specific DNA sequence. In PCR reactions, primers are used to guide DNA polymerases to replicate target DNA (Naumovski & Friedberg, 1984; Weissenmayer et al., 2002). The melting temperature (Tm value) for primer binding is typically set between 50–65 °C to ensure that the primers can anneal specifically to the target DNA. The Tm value is closely related to the GC content of the primer sequence, and normally the GC content should be 40–60%. When there are local regions with too low or too high GC content, it is difficult to design subcloning primers and the PCR amplification may not go well or may even end in failure (Green & Sambrook, 2019; Li et al., 2011; Strien, Sanft & Mall, 2013).

In order to solve the above problems encountered in the experimental process, a stand-alone software with a graphical interface was written in Python, with which users can check and adjust the GC content of the codon-optimized sequence to eliminate the regions with too high or too low local GC content within the gene sequence. Working in a manner of dynamic display, the software not only provides a panoramic view of GC content but also allows real-time adjustments of local GC content via manual change of adjacent synonymous codons. This dual functionality is useful for the downstream analysis and application of the results obtained from the current codon optimization tools. This feature facilitates better primer design and gene synthesis, potentially reducing experimental failures related to suboptimal GC content.

Materials and Methods

Computer hardware and software

Portions of this text were previously published as part of a preprint (Lin et al., 2024). The software, which is called visual_codon.py, can be run on a common desktop or laptop computer with Windows, MacOS, or Linux installed. It is a free, open-source Python program written in an object-oriented programming style, and the detailed annotation is provided. These make our program easy to understand and use.

The program was developed using Python 3.12 (https://www.python.org) and matplotlib 3.8.2 (Barrett et al., 2004), with which our program works. However, other versions of Python and matplotlib may work as well. European Molecular Biology Open Software Suite (EMBOSS) is also required for our program (Rice, Longden & Bleasby, 2000). The codon usage frequency tables of E. coli and other species are obtained from the Genscript website (https://www.genscript.com/tools/codon-frequency-table). The example gene used in this study is the dnaN from Mycobacterium tuberculosis (TBdnaN) with GeneID 887092 (Cole et al., 1998; Gui et al., 2011). The files loaded by the software include the original sequence TBdnaN.fasta, and the derivatives from multiple codon optimization tools. The program and the example files are stored in GitHub at https://github.com/shiqiang-lin/visual_codon. We provided a tutorial named tutorial.txt for running the program in the above GitHub link and the results are analyzed later in the Results section.

Flow chart of the program

The flow chart of the program is shown in Fig. 1. The parts of the program include the tabular display of codon information, the replacement of codons, and the GC content plot of codon sites, described as follows.

Figure 1 Flowchart of program.

InitialDialog: Launch the initial screen of the program; ‘start main GUI’: launch the program and open the main interface; customize_organism(): customize the codon table usage of host species; check_table_content(): check whether the content filled in meets the requirements; save_to_file(): save the codon table usage of host species to a txt file; ‘Open Gene 1 to edit’: open a FASTA gene file; read_sequence_from_file(): read a sequence file; ‘validity check’: check the validity of the sequence file; insert_itmes(): insert an item to the Treeview; update_gc_graph(): plot the GC content of the gene. update_selected_item(): change the codon of the selected amino acid; save_optimized_gene(): save optimized gene sequence to a FASTA file; export_table_to_txt(): export the Treeview table to a txt file; export_changed_codons_to_txt(): export the changed codons to a txt file; ‘Import Gene 2 to compare’: import a fasta or fa gene file, which codes the same protein as ‘Open Gene 1 to edit’. The program’s mode will become read-only.

After the user opens the gene sequence file (Gene 1), the program will extract the gene sequence information from it. It is worth mentioning that the program may be used in two major ways, one being a read-only/comparison mode and the other being an adjustment/optimization mode. Here, we are dealing with the read-only/comparison branch. Therefore, the user needs to import the codon-optimized gene sequence file (Gene 2), which will be displayed in the tkinter’s Treeview. While the program is running, the user will be able to see two entries showing the original and optimized gene sequences. This interface design allows users to view and compare gene sequence information optimized by different web services or codon optimization software. The mode is for comparison and is ReadOnly.

If the user decides to manually optimize the gene sequence, then reopen the gene sequence to be optimized (Gene 1). According to the GC content plot at the bottom of the Graphical User Interface (GUI), the user can find the local high and low points, and manually adjust the codons as needed. The effect of codon change on the local GC content is real-time, which provide a good experience of program operation.

Users can export the optimized gene sequence for subsequent experimental analysis. Users can also export the entire Treeview table and copy it to Excel or Numbers for statistical analysis.

When calculating the GC content of each codon position, each codon itself and the three adjacent codons before and after it, a total of 7 codons, i.e., 21 bases will be included in the calculation. Because the loci of the head and tail codons do not meet the above rules, the GC content of the fourth codon and the codon fourth from the bottom can be used as references, respectively. The results are displayed in the Treeview control of tkinter and plotted with matplotlib at the bottom of the GUI interface.

Results

Verification of GC content before and after gene sequence optimization

After running the software, the gene sequences before and after optimization can be loaded successively, and the local GC contents before and after sequence optimization can be compared. As shown in Fig. 2, the TBdnaN sequence is processed by four different codon optimization tools, respectively. The GC contents of the original sequence and the optimized sequence are visualized with lines of brown or blue so that users have a more intuitive understanding of the sequence information before and after optimization.

Figure 2 The GC content of TBdnaN before and after optimization by four codon optimization tools.

(A) Genscript tool. (B) IDT tool. (C) ATGme. (D) CodonWizard. The brown and blue lines represent the GC contents of sequences before and after optimization, respectively.

It can be seen from the figure that after TBdnaN optimization, due to the different considerations of different codon optimization tools for codon optimization parameters such as codon adaptation index (CAI) (Sharp & Li, 1987) and codon pair score (CPS) (Coleman et al., 2008), there are still local highs or lows of GC content.

Adjustment of local GC content of gene sequence

After the sequence check before and after the optimization in the previous step, the user can adjust the GC content of the optimized sequence where there are too high or too low locals. The middle of the software graphical interface shows the codon that can be replaced and the frequency of codon usage. The user can change the codon according to their own needs, and the modified results will be visually displayed in the Treeview table at the top of the interface and the GC content plot at the bottom of the interface, as shown in Fig. 3.

Figure 3 The results of adjusting the local high and low points of GC content.

The brown and blue lines represent the GC contents before and after manual optimization, respectively. The bold ‘a’ stands for the peak and bold ‘b’ for the valley.

Comparing the GC contents at the bottom of the interface before and after adjustment, it can be found in Fig. 3 that the GC content corresponding to the peak ‘a’ appears at the position of the 6th amino acid, and the GC content value corresponding to the peak decreases from 85.71% to 61.9% after replacing the codons at the 4th–7th positions. Based on the original GC content (brown horizontal line) of the sequence in the GC content graph at the bottom of the GUI interface, the difference between the peak and the mean GC content is reduced by more than 20. After replacing the codons at the 282nd and 284th positions, the GC content value of the valley ‘b’ increased from 23.81% to 33.33%, and the difference between the valley and the mean GC content was reduced by more than 10. These can be visualized from the GC content plot, or the user can drag the scroll bar from the Treeview table to the corresponding codon position to see the exact value before and after changing the codons.

In addition, users can export the modified sequence as a fasta file, or export the Treeview table as a txt file. As shown in Fig. 4, the fasta file is the sequence that has been further optimized by our software, and the txt file contains the details before and after the optimization of each codon locus.

Figure 4 Exporting the program running results.

(A) Screenshot of the files produced by the program. (B) Screenshot of the fasta file saved. (C) Screenshot of exported txt file (partial). (D) Screenshot of changed codons txt file.

It is worth mentioning that we use a codon dictionary in the program to ensure that the exported fasta sequence is consistent with the gene sequence opened by the program in terms of the protein sequence. Users can also use alignment software such as EMBOSS needle (Ionescu, 2019; Koyama, Platt & Parida, 2020) to compare the exported sequence with the original sequence for confirmation.

Discussion

This study provides a tool for GC content verification and adjustment for the optimized gene sequence to ensure that the GC content of gene sequence is relatively consistent. By aligning the original gene sequence with the optimized sequence, users can detect and correct the sites that may be problematic in subsequent experiments. The software provides visual alignment and detailed result export functions to help researchers ensure that the optimized gene sequence meets the design requirements of the subsequent experiments such as subcloning and gene synthesis via PCR.

We provide the source code and add detailed comments, which improves the readability of the code and enables users to better understand the logic and functionality. When code needs to be modified or maintained, comments can provide relevant contextual information to improve productivity. At the same time, because the program provides the source code, the program is not only limited to the application of E. coli codon optimization. Users can customize it for other species according to their own needs.

The program provides a graphical interface that allows users to interact with the program in an intuitive and user-friendly way, thus improving the user experience. Through the graphical interface, users can operate through mouse clicks, without the need to memorize and enter command line parameters. For example, the change of the GC content is displayed by the change of color, so that the user can easily understand the function and operation process of the program. The implementation of the GUI interface of this program makes use of the tkinter library, which is a standard GUI library for Python. It is integrated directly in Python and there is no need to install. Meanwhile, tkinter can run smoothly on multiple operating systems, including Windows, Mac, and Linux, which is a powerful and easy-to-use GUI library (Aires-de-Sousa, 2024; Chauhan et al., 2023; Garcia et al., 2019; Shaikh et al., 2008).

Our program has prepared a table of codon usage frequencies for more than a dozen species. Codon frequency tables for other species are available from the Genscript website (https://www.genscript.com/tools/codon-frequency-table) or the Codon Usage Database (http://www.kazusa.or.jp/codon/). If the user’s research species is outside of these species, then we also provide the functionality to customize the codon usage frequency table.

The program has several limitations. It can only optimize the coding region and adjust the local GC content by changing the codons. If the gene is in antisense strand, then the user needs to complement reverse the sequence using EMBOSS revseq or other tools. After obtaining the sense strand, the user can optimize it with the commonly used optimization software, and then use our program to check the GC content of the obtained sequence, and decide whether to modify the synonymous codons to adjust the local GC content according to the specific situations.

Codon optimization is limited by a variety of conditions. It is not easy to meet various constraints in practice. Our program can view and compare the GC content of gene sequences, manually modify the synonymous codons to change the local GC content where it is too low or too high. However, these modifications may bring new problems. Therefore, after the modification, various checks need to be done to ensure that no new problems appear, such as inappropriate restriction sites (check with EMBOSS restrict), direct duplication (check with EMBOSS equicktandem), palindrome (check with EMBOSS palindrome), and reverse duplication (check with EMBOSS einverted) (Rice, Longden & Bleasby, 2000). These commands have been integrated to our program, which may help users check the gene sequences conveniently.

Conclusions

With our program, users can check the GC content of the optimized gene sequence and adjust the optimization according to their own needs, and the modified results can be visualized. The software assists in gene cloning and its functional studies.

Additional Information and Declarations

Competing Interests

Author Contributions

Data Availability

The authors declare that they have no competing interests.

Shiming Lin performed the experiments, analyzed the data, prepared figures and/or tables, authored or reviewed drafts of the article, and approved the final draft.

Fei Xu performed the experiments, analyzed the data, prepared figures and/or tables, authored or reviewed drafts of the article, and approved the final draft.

Bifang Huang performed the experiments, prepared figures and/or tables, and approved the final draft.

Li-li Zhao performed the experiments, prepared figures and/or tables, and approved the final draft.

Danni Pan performed the experiments, prepared figures and/or tables, and approved the final draft.

Shiqiang Lin conceived and designed the experiments, performed the experiments, analyzed the data, prepared figures and/or tables, authored or reviewed drafts of the article, and approved the final draft.

The following information was supplied regarding data availability:

The program source code and gene sequence files are available at GitHub and Zenodo:

- https://github.com/shiqiang-lin/visual_codon

- shiqiang-lin. (2024). shiqiang-lin/visual_codon: 1.2.1 (1.2.1). Zenodo. https://doi.org/10.5281/zenodo.14249496.

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
