# Peer review of "Visual codon: a user-friendly Python program for viewing and optimizing gene GC content"

_PeerJ, doi:10.7717/peerj.18755_

## Round 0.1 · original submission · Major Revisions

All the reviewers' concerns must be addressed satisfactorily for the manuscript to be considered for publication

Reviewer 1 ·

Basic reporting

Manuscript submitted here is largely similar (if not the same) to the version I reviewed 3-4 months ago. While I appreciate the authors for adding documentation to their github repo, my major concerns from the earlier review still remains. I'm copy/pasting the major concern from my earlier review:

"I do not find the tool in the current form to be of substantial effort to benefit journal publication. Most of the manuscript deals with how to run the tool and interpret the output, which should be part of the tool documentation and not instead in the form of a journal article. Tool updates in the future could result in the documentation becoming out of date and therefore unreliable to the user."

Also as I stated in the earlier review, supplementary material is mentioned in line 120, but they were not available to me for review.

Experimental design

NA

Validity of the findings

NA

Reviewer 2 ·

Basic reporting

The manuscript is clear and unambiguous and the revised version has addressed previous comments.

Experimental design

The revised version has presented the research question well defined and meaningful.

Validity of the findings

The revised manuscript has assessed impacts and novelty.

·

Basic reporting

The authors present an interface where the users can modify codons while monitoring the local GC%. I am familiar with this optimization problem and agree with the need for tools. The software works well, is easy to install, and is intuitive enough that I could use it after ~20s of exploration, without reading any documentation - kudos to the authors.

Even though it is not the review’s goal to assess impact and novelty, let’s point that the tool’s scope is relatively narrow:

- It will only work for sequences representing exactly a gene, not sequences containing other elements (promoters, UTRs, etc.). And it won’t work if the gene is on the antisense strand.

- It only supports E. coli codon frequencies and no other organism, and these frequencies are hard-coded within the app. The authors claim this is easy to change by editing the software, but this still places a burden on the user.

- It doesn’t display any other sequence properties impacting cloning, such as restriction sites, direct repeats, etc. these features just as important as GC%, and if fixing a sequence’s GC% leads to issues with these other features then the optimization will have been destructive.

The above remarks matter because pre-existing tools already address these needs, and are not mentioned in the manuscript. First, the popular Geneious Prime has a GC content viewing feature that is integrated with translated protein visualization, sequence annotation visualization, enzyme restriction site visualization etc. (see [1] for a public screenshot). Another popular sequence editor, Benchling, also enables to fix local GC% by juggling codons, with a very integrated sequence visualization tool. Second, the authors claim to address “a significant gap in current codon optimization tools, which often overlook local GC content variations”. However, codon optimization tools such as DnaChisel, GeneOptimizer, JBEI-BOOST, Benchling’s optimizer, BaseBuddy, all offer options to fix local GC%. And these tools support many organisms, optimization for restriction site and sequence repeat avoidance, support sequences that are not only CDSs, and many other features not supported by `visual_codons`. Therefore, rather than simply suggesting the use of `visual_codon`, the authors should at least mention current alternatives and explain when a user would rather `visual_codon` rather than their tool.

In conclusion, while the software does work and is intuitive and does bring a way of working with sequences that is not fully covered by any existing tool, it should better present pre-existing software that address local GC content tuning, so a reader would understand when to use this tool in particular.

Experimental design

NA (no experiments)

Validity of the findings

NA (no findings)

Additional comments

As a minor remark, the tool could rely more on existing packages from the python ecosystem. For instance, biopython provides fasta parsing (no need for all the fasta-parsing text), and python-codon-optimization gives codon frequency tables for hundreds of organisms (no need to hardcode a frequency table).

Reviewer 4 ·

Basic reporting

No major comments. The article is generally written in clear professional English. The work is appropriately supported with references to the literature. The figures are of a suitable quality for readers.

Experimental design

This work fits the aims and scope of the journal under “Bioinformatics Software Tools”. The need fulfilled by the tool (visual_codon) is relevant and clearly stated. Sufficient information is provided to assess and use the software.

Validity of the findings

All relevant underlying data and code are provided. The simple conclusion that this tool may actually aid the design of gene constructs for heterologous expression following using another codon-optimization software is clearly supported.

Additional comments

If I had to design a construct for heterologous expression, I would probably use this tool in that process.

Although generally well written, the manuscript could benefit from some optimizations and clarifications.
Line 20: I suggest breaking this long sentence into two sentences ‘tools, however’ => ‘tools. However’
Line 23: “based on Python language” => written in python
Line 25: E. coli is only the default. It would be good if this could be clear as soon as possible as a worker using any other heterologous expression host may be turned off at this point.
Lines 39-41: the “such as Thermo ….” should come after “commercial companies”
Line 51: missing an article “…. amplify a specific ….”: or otherwise rephrase this sentence.
Line 52: “instruct” => “guide”
Line 53: The ‘m’ in Tm value should be subscripted. There is at least one other instance of this throughout the article. Please fix all of them.
Line 54: I suggest replacing “degrees Celsius” with its unit symbol “°C”
Line 55: Delete “level of”
Line 57: Rephrase “too little or too much GC content”. e.g too low or too high GC content
Line 61: “is written” to “was written”
Line 61: “in Python language” to “in python”
Line 63: “WYSIWYG” although I understood this easily, I’m not sure that it’s an appropriate abbreviation. Perhaps this sentence should be rephrased.
Lines 63-64: I think “the software not only optimizes codon usage” is a little misleading. It helps a user to manually optimize things and seems to be designed to run after a separate codon optimization tools has already been used. Consider rephrasing this.
Line 70: The word “hardware” is in the title but no mention of hardware requirements is made. It is obvious that this tool would run on very modest hardware and it would be a good service to the reader to state this.
Line 71: “which is visual_codon.py” => which is called visual_codon.py
Line 73: delete “for the users”
Line 77: Requirements are stated for python=3.12 and matplotlib=3.8.2. I don’t see how either of these could be so strict from reading the source code and I was able to run the software on older versions of python (3.8) and matplotlib (3.6.0). Could the authors reconsider this statement (based on their own more extensive testing) - perhaps it may be more appropriate to state that the software was developed with python3.12 and matplotlib3.8.2 but that other versions of these may also work?
Lines 78-79: I think it is unnecessary detail to mention the development environment used.
Line 79: If keeping mention of the fact that IDLE was used, I would still advocate that “that comes with Python 3.12” be deleted as IDLE is separate from python - and python can be used without it.
Line 82: “run” to “loaded” In this context, I would reserve “run” to refer to executable files.
Lines 82-88: I would suggest rephrasing this section. Instead of listing file names, I would indicate that the github repository includes an example based on the TBdnaN sequence and derivatives from multiple codon optimization tools.
Line 96: “Next, the user needs to….” I think some confusion arrives here. It needs to be clear by this point that the program may be used in two major ways, one being a read-only comparison mode and the other being an adjustment / optimization mode, and that this paragraph is dealing with the read-only / comparison branch.
Line 102: delete “The program stipulates that”
Lines 102-107: This paragraph seems to be out of place. This section (Flow chart of the program) should focus on describing the two major modes of the program, the read-only comparison mode, and the optimization / edit mode. I think the information about how local GC content is calculated should be presented elsewhere.
Line 132: ‘File - Open’ seems like it refers to a different version of the software to the V1.1 that is currently out on github. I see this option as ‘File - Open Gene 1 to edit’.
Lines 132-133: “File - Import optimized gene” appears as “File - Import Gene 2 to compare” when I run the software.
Line 145: ‘File - Open’ appears as ‘File - Open Gene 1 to edit’ when I run the software.
Lines 210-212: As I mentioned earlier (in the abstract), I think it would be a service to the reader if this point could be more up-front.
Lines 213-214: “in a more intuitive and friendly way” => in an intuitive and user-friendly way
Lines 220-221: I think the sentence about third-party libraries for tkinter is irrelevant to this work. The authors could consider removing it.
Line 230: “Lines 57 - 90 of the program”: For the program to be user-friendly, it would be better if changing the optimal codon frequency could be done in a way where the user does not need to edit the source code.

Comments about the software:
1. There are a mixture of f-string style and ‘%’ style strings used. This isn’t wrong but it could be nice to use one style or the other consistently.
2. The “Set graph X axis” function (under the “Edit”) menu is a confusing name. Perhaps “adjust graph font” or something similar might be a better substitute

·

Basic reporting

No comment

Experimental design

The code has been improved after the review by authors. The program has now been converted using object oriented approach which will improve the efficiency and reliability of the software.

Validity of the findings

No comments

Additional comments

As suggested earlier, the authors have added a detailed tutorial on the usage of the software that can be found at https://github.com/shiqiang-lin/visual_codon

---

## Round 0.2 · Minor Revisions

All of the reviewers' criticism should be addressed in the revised version.

Reviewer 1 ·

Basic reporting

* Unfortunately, I continue to find the manuscript too narrow in scope. Authors developed a tool, which assists the user in manually identifying and then modifying the codons in an already optimized sequence in an effort to reduce possible high/low GC related problems during cloning and expression. Had the tool also been capable of addressing some of the tool limitations rightly mentioned by the authors (such as inappropriate restriction sites, direct duplication, etc.), I would have found its scope to be more appropriate.

* Much of the text in "materials and methods" outlines the steps to use the tool, which I believe should be part of the tool documentation and not just as a part of the publication.

* Line 23 in abstract ".. we present a stand-alone software written in Python, which can check and adjust the GC content of sequence-optimized genes..." as well as line 64 in introduction ".. which can check and adjust the GC content of the codon-optimized sequence..." appear to imply that their tool can automatically identify and adjust the GC content. However, this task is instead a manual process performed by the user.

* As per PeerJ's Data and Materials Sharing, source code available via Github here needs to be archived at Zenodo (or similar), which in turn assigns a DOI. This is for reproducible data sharing purposes.

* Line 74 cites preprint version (Lin et al, 2024) of the manuscript under review here, which is an uncommon practice. Such self-referencing practices should be discouraged.

* Tools mentioned in Line 40 need to be cited. Tool name Genscipt shoud instead be Genscript.

Experimental design

NA

Validity of the findings

NA

Additional comments

NA

·

Basic reporting

I appreciate the new mention of pre-existing tools that can also address local GC content optimization, DnaChisel (with the conflict of interest that I am the author), GeneOptimizer, and BaseBuddy, However, instead as presenting these tools as previous art in GC content optimization, the authors on the contrary mis-characterize their features: they write "The sequence optimized with the above tools may still have some regions where the GC contents are too high or too low". This is false, DnaChisel GeneOptimizertools and BaseBuddy all let the user set bounds on local GC content (over a given window size) and optimize the sequence according to these constraints. It is fine to put forward Visual Codon's interface and highlight its interactivity and it's ability to tweak sequences nucleotide by nucleotide. But it is not fine to misrepresent other tools in a way that could lead a reader to avoid using these even though they could be a solution to their problems.

Other than this important issue, I believe my remarks have been addressed and I particularly appreciate that the authors have improved parts of the software, in particular by providing a choice of multiple species.

Experimental design

NA

Validity of the findings

NA

Additional comments

NA

Reviewer 4 ·

Basic reporting

The article is generally written in clear professional English. The work is appropriately supported with references to the literature. The figures are of a suitable quality for readers. Data required to reproduce the authors’ results are easily available.

Experimental design

This work fits the aims and scope of the journal under “Bioinformatics Software Tools”. The need fulfilled by the tool (visual_codon) is relevant and clearly stated. Sufficient information is provided to assess and use the software.

Validity of the findings

All relevant underlying data and code are provided. The simple conclusion that this tool may actually aid the design of gene constructs for heterologous expression following using another codon-optimization software is clearly supported.

Additional comments

I previously reviewed and earlier version of this manuscript. I am satisified with the manner in which the authors addressed the various minor comments I made. Having reviewed the software again, it is a good improvement that they made it much easier to load codon tables for other expression hosts.

---

## Round 0.3 · accepted · Accept

All the it this the manuscript satisfactory for publication.